# AF-UMC: An Alignment-Free Fusion Framework for Unaligned Multi-View Clustering

**Bohang Sun**[1,2]**, Yuena Lin**[1]**, Tao Yang**[3]**, Zhen Zhu**[4,5]**, Zhen Yang**[1]**, Gengyu Lyu**[1]*

[1]College of Computer Science, Beijing University of Technology
[2]Engineering Research Center of Intelligent Perception and Autonomous Control, Ministry of Education
[3]Idealism Beijing Technology Co., Ltd.
[4]School of Computer Science and Technology, Zhejiang Sci-tech University, China
[5]KEYI College, Zhejiang Sci-tech University, China
sunbohang@emails.bjut.edu.cn, yuenalin@126.com, yangtao@ilxzy.cn
hzzhuzhen@yeah.net, yangzhen@bjut.edu.cn, lyugengyu@gmail.com

## Abstract

The Unaligned Multi-view Clustering (UMC) aims to learn a discriminative cluster structure from unaligned multi-view data, where the features of samples are not completely aligned across multiple views. Most existing methods usually prioritize employing various alignment strategies to align sample representations across views and then conduct cross-view fusion on aligned representations for subsequent clustering. However, *due to the heterogeneity of representations across different views, these alignment strategies often fail to achieve ideal view-alignment results, inevitably leading to unreliable alignment-based fusion.* To address this issue, we propose an alignment-free consistency fusion framework named AF-UMC, which bypasses the traditional view-alignment operation and directly extracts consistent representations from each view to perform global cross-view consistency fusion. Specifically, we first construct a cross-view consistent basis space by a cross-view reconstruction loss and a designed Structural Clarity Regularization (SCR), where autoencoders extract consistent representations from each view through projecting view-specific data to the constructed basis space. Afterwards, these extracted representations are globally pulled together for further cross-view fusion according to a designed Instance Global Contrastive Fusion (IGCF). Compared with previous methods, AF-UMC directly extracts consistent representations from each view for global fusion instead of alignment for fusion, which significantly mitigates the degraded fusion performance caused by undesired view-alignment results while greatly reducing algorithm complexity and enhancing its efficiency. Extensive experiments on various datasets demonstrate that our AF-UMC exhibits superior performance against other state-of-the-art methods.

## 1 Introduction

Multi-view data is usually collected from multiple sources, which is represented by several heterogeneous features. For instance, in personalized online recommendations, diversified individual preferences are collected from various e-commerce platforms. To recommend reliable products, it is necessary to comprehensively integrate all these preferences. However, in practical scenarios, the collected multi-view preference data is usually unaligned since different platforms do not store data in a unified order in general. Under such conditions, the traditional multi-view learning methods lose their capability to fuse the unaligned multi-view data [38, 37].

---

*Gengyu Lyu is the corresponding author.

39th Conference on Neural Information Processing Systems (NeurIPS 2025).

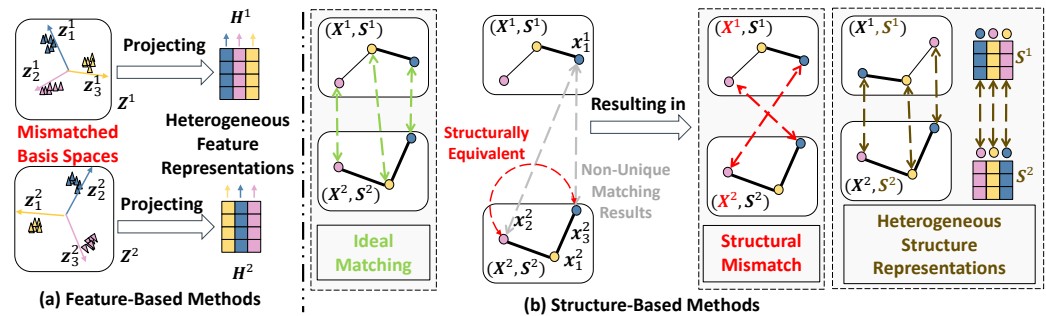

(a) Feature-Based Methods

(b) Structure-Based Methods

Figure 1: The drawbacks of existing feature-based and structure-based methods. (a) The unintended mismatched basis spaces of feature-based methods. Triangles in different colors denote samples of different categories, and solid arrows in different colors denote different basis vectors $\{\mathbf{z}_i^v\}_{i=1}^c$. Since $\mathbf{Z}^1$ and $\mathbf{Z}^2$ are independently constructed in each view without any cross-view constraints, the correspondences between basis vectors $\{(\mathbf{z}_i^1, \mathbf{z}_i^2) | 1 \le i \le c\}$ are often incorrect across views (i.e., $\mathbf{Z}^1$ and $\mathbf{Z}^2$ are often mismatched across views), which induces the heterogeneity of projected feature representations ($\mathbf{H}^1$, $\mathbf{H}^2$) across views. (b) The undesired structural equivalence across basis vectors of structure-based methods. Circles in different colors denote different basis vectors $\{\mathbf{x}_i^v\}_{i=1}^c$ (i.e., the nodes on structure $\mathbf{S}^v$), and thicker black lines indicate a closer structural relationship between the basis vectors. $\mathbf{x}_2^2$ and $\mathbf{x}_3^2$ are structurally equivalent when they have the coincident structural correspondence on $\mathbf{S}^2$ (i.e., they share the coincident neighbor node $\mathbf{x}_1^2$ and have the coincident relationship with $\mathbf{x}_1^2$) [39]. When $\mathbf{x}_1^1$ tries to match the corresponding basis vector $\mathbf{x}_3^2$ on $\mathbf{S}^2$ through structural correspondence, it will find two candidates ($\mathbf{x}_2^2$, $\mathbf{x}_3^2$) that have the coincident structural correspondence on $\mathbf{S}^2$, which induces non-unique matching results. This easily leads to structural mismatched basis spaces ($\mathbf{X}^1$, $\mathbf{X}^2$) and heterogeneous structure representations ($\mathbf{S}^1$, $\mathbf{S}^2$).

The key to learn from unaligned multi-view data lies in how to fuse cross-view information with the unaligned sample features across different views. Existing unaligned multi-view clustering (UMC) methods, mainly divided into feature-based and structure-based, provide an effective solution, which prioritizes employing various alignment strategies on sample representations and then fuses these aligned representations into a cross-view consistent representation for clustering. For instance, feature-based UMC methods [35, 3, 8], construct an orthogonal matrix $\mathbf{Z}^v \in \mathbb{R}^{c \times D_v}$ to represent a basis space with $c$ basis vectors $\{\mathbf{z}_i^v\}_{i=1}^c$ in each view $v$, and obtain $c$-dimensional feature representation $\mathbf{H}^v \in \mathbb{R}^{N \times c}$ by projecting samples $\mathbf{X}^v \in \mathbb{R}^{N \times D_v}$ onto $\mathbf{Z}^v$, formally expressed as $\|\mathbf{X}^v - \mathbf{H}^v\mathbf{Z}^v\|_F^2$. After that, $\mathbf{H}^v$ is used for cross-view representation alignment through various alignment strategies, such as introducing a learnable alignment matrix with $\mathcal{O}(N^2)$ memory or Hungarian algorithm, and then the aligned feature representations are used to fuse the cross-view consistent representation $\mathbf{H}^*$:

$$\min \sum_{v=1}^V \|\mathbf{X}^v - \mathbf{H}^v\mathbf{Z}^v\|_F^2 + \sum_{v=1}^V \|\mathbf{H}^* - \Phi(\mathbf{H}^v)\|_F^2,$$
$$s.t. \ \forall v, \mathbf{Z}^v(\mathbf{Z}^v)^T = \mathbf{I}. \tag{1}$$

In Eq. (1), $\Phi(\cdot)$ indicates the associated alignment strategies, $V$ and $N$ are the number of views and samples, respectively. However, the constructed basis spaces $\{\mathbf{Z}^v\}_{v=1}^V$ are often mismatched across views due to the independent construction process $\|\mathbf{X}^v - \mathbf{H}^v\mathbf{Z}^v\|_F^2$ without any cross-view constraints, as shown in Figure 1 (a), which induces the heterogeneity of projected representations $\{\mathbf{H}^v\}_{v=1}^V$, and the heterogeneity makes trouble for subsequent alignment strategies to achieve ideal view-alignment results, leading to unreliable cross-view fusion. For structure-based methods [18, 40, 32], they project structure representation $\mathbf{S}^v$ by a self-representation term $\|\mathbf{X}^v - \mathbf{S}^v\mathbf{X}^v\|_F^2$, which is an analogue of the projection operation $\|\mathbf{X}^v - \mathbf{H}^v\mathbf{Z}^v\|_F^2$ in feature-based methods. Consequently, each row vector of $\mathbf{X}^v$ can be regarded as both a sample feature and a basis vector, and the obtained structure representation $\mathbf{S}^v$ for $\mathbf{X}^v$ indicates the structure of both samples and basis vectors. In this case, their alignment strategies $\Phi(\cdot)$ on $\{\mathbf{S}^v\}_{v=1}^V$ simultaneously perform both multi-view sample alignment and basis space matching, and then the aligned sample structures $\{\Phi(\mathbf{S}^v)\}_{v=1}^V$ are used to fuse the

cross-view consistent structure $\mathbf{S}^*$:

$$\min \sum_{v=1}^{V} \|\mathbf{X}^v - \mathbf{S}^v \mathbf{X}^v\|_F^2 + \sum_{v=1}^{V} \|\mathbf{S}^* - \mathit{\Phi}(\mathbf{S}^v)\|_F^2. \qquad (2)$$

Nevertheless, since the basis space $\mathbf{X}^v$ is directly constructed using view-specific sample features, the view-specific inherent heterogeneity is completely reserved and induces the heterogeneity for structures $\{\mathbf{S}^v\}_{v=1}^{V}$ across different views, where the alignment strategies on $\{\mathbf{S}^v\}_{v=1}^{V}$ also lose their expected capability. In addition, it is difficult to obtain ideally matched basis spaces through structural match, since structurally equivalent basis vectors disrupt the ideal matching results, as shown in Figure 1 (b), which also induces the heterogeneity of representations $\{\mathbf{S}^v\}_{v=1}^{V}$, deceiving the alignment and the subsequent multi-view fusion towards a biased direction. To sum up, the current methods, whether feature-based or structure-based, suffer from the common limitation: **Their cross-view fusion operation depends on aligned representations, but the ideal view-alignment results often fail to be obtained by alignment strategies due to the heterogeneity of representations across views, inevitably inducing unreliable alignment-based fusion.**

To address the above issues, in this paper, we propose an alignment-free consistency fusion framework AF-UMC for unaligned multi-view clustering, which directly extracts consistent representations from each view for globally fusing a cross-view consistent representation and does not require additional alignment strategies. Specifically, we first construct a cross-view consistent basis space. On one hand, the basis space is designed to capture cross-view shared information from multiple views, where exclusive diversity is filtered out and the shared consistency is reserved. On the other hand, a Structural Clarity Regularization (SCR) is designed to prevent the basis space from learning structurally equivalent basis vectors and to encourage the basis space to capture matched information from different views. Afterwards, autoencoders are employed to extract consistent representations from each view by projecting view-specific data to the constructed basis space. Finally, these extracted representations are globally pulled together for further fusing a cross-view consistent representation by a designed Instance Global Contrastive Fusion (IGCF), and then the clustering results are obtained by K-means clustering. During the whole process, different from previous methods that utilize alignment strategies with $\mathcal{O}(N^2)$ complexity to align representations for cross-view fusion, AF-UMC directly extracts consistent representations from each view for global fusion, avoiding the additional cost of alignment strategies while mitigating the risk of fusing non-corresponding representations. In summary, the main contributions of this paper lie in:

- We analyze a common problem in existing unaligned multi-view clustering methods: alignment strategies often fail to achieve ideal view-alignment results due to the inherent heterogeneity of representations across different views, inevitably leading their alignment-based cross-view fusion toward a biased direction.

- We propose an alignment-free consistency fusion framework AF-UMC, which does not require additional alignment strategies and directly extracts consistent representations from each view by projecting view-specific data to a constructed cross-view consistent basis space, and then globally fuses them into a cross-view consistent representation.

- Extensive experimental results on various datasets demonstrate that our proposed model exhibits superior performance against other state-of-the-art algorithms. Moreover, we conduct comprehensive ablation studies on both loss functions and model components, clearly demonstrating their effectiveness within our AF-UMC.

## 2   Related works

**Multi-view clustering.**   Multi-view clustering aims to unsupervisedly fuse multi-view data to differentiate crucial clusters, and is a fundamental task in the fields of data mining [20, 44, 30, 31, 25, 2, 12], pattern recognition [19, 29, 41, 13, 9, 6, 28], etc. The key to dealing with such a problem lies in how to fuse cross-view information and obtain a consistent representation for clustering. Current multi-view clustering methods are mainly divided into two categories, i.e., shallow methods and deep learning-based methods. For instance, Wu et al. [34] propose a shallow method, which integrates multi-view samples into a unified tensor through matrix factorization and then utilizes a low-rank kernel tensor constraint to fuse cross-view consistent representation. Wang et al. [27] propose a deep learning-based method, which employs graph autoencoders to pull together structurally similar samples and then introduces contrastive learning for fusing cross-view consistent representation.

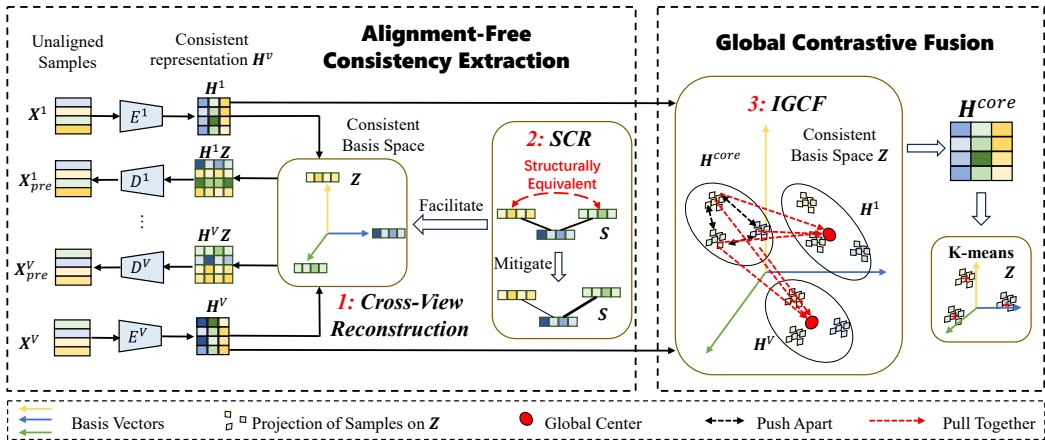

Figure 2: The overview of AF-UMC, which consists of two main stages: alignment-free consistency extraction and global contrastive fusion. In the first stage, we construct a cross-view consistent basis space through a cross-view reconstruction and a designed Structural Clarity Regularization (SCR), where autoencoders extract consistent representations from each view through projecting view-specific data to the constructed basis space. In the second stage, these extracted representations are globally pulled together for fusing a cross-view consistent representation $\mathbf{H}^{core}$ according to a designed Instance Global Contrastive Fusion (IGCF), and then the final clustering results are obtained by K-means clustering.

**Unaligned multi-view clustering.** Unaligned multi-view clustering aims to cluster the multi-view data where the sample features are not completely aligned across views [35, 3, 8, 18, 40, 32, 21]. The key to dealing with such a problem lies in how to fuse cross-view information for clustering under the unaligned sample features. Existing methods can be divided into two main categories, i.e., feature-based methods and structure-based methods. For instance, Ji et al. [8] propose a feature-based method, which first introduces a learnable alignment matrix with $\mathcal{O}(N^2)$ memory to align multi-view feature representations, and then utilizes a low-rank kernel tensor constraint to capture cross-view consistency while fusing a cross-view consistent feature representation. Xin et al. [35] also propose a feature-based method, which employs the Hungarian algorithm to align multi-view feature representations and then introduces a cross-view contrastive loss to pull together cross-view positive representations for fusing a cross-view consistent feature representation. Wen et al. [32] propose a structure-based method, which first extracts a structure representation by a self-representation function in each view and introduces a learnable alignment matrix with $\mathcal{O}(N^2)$ memory to structurally align cross-view samples, and then introduces a low-rank kernel constraint to fuse aligned sample structures into a cross-view consistent structure. Although these methods have achieved competitive performance, they still suffer from the same drawback: Their alignment strategies often fail to achieve ideal view-alignment results due to the heterogeneity of representations across different views, inevitably leading to unreliable cross-view fusion.

## 3   Method

In this paper, we propose an alignment-free consistency fusion framework AF-UMC for unaligned multi-view clustering, which eliminates the requirement for alignment strategies and directly extracts consistent representations from each view to perform global cross-view consistency fusion. The AF-UMC is decomposed into two stages: *Alignment-Free Consistency Extraction* and *Global Contrastive Fusion*. In the first stage, we extract consistent representations from each view by projecting view-specific samples to the constructed cross-view consistent basis space. In the second stage, these extracted representations are globally pulled together to fuse the cross-view consistent representation for clustering. Figure 2 illustrates the overview of our proposed method.

## 3.1 Problem definition

Given an unaligned multi-view dataset $\mathbf{X} = \{\mathbf{X}^v\}_{v=1}^V$ with $V$ views, where $\mathbf{X}^v = \{\mathbf{x}_i^v\}_{i=1}^N \in \mathbb{R}^{N \times D_v}$ denotes the sample features of $v$-th view, $D_v$ is the dimension of $\mathbf{X}^v$ and $N$ is the number of samples. The goal of AF-UMC is to fuse multi-view information for separating multi-view data to pre-define $k$ clusters. Notably, the multi-view sample features cannot be directly fused since $\{(\mathbf{x}_i^p, \mathbf{x}_i^q), p \neq q\}$ are often derived from different samples in the unaligned multi-view dataset.

## 3.2 Alignment-free consistency extraction

In this stage, we construct a cross-view consistent basis space $\mathbf{Z} = \{\mathbf{z}_i\}_{i=1}^c \in \mathbb{R}^{c \times d}$ to capture cross-view consistency from multiple views and extract consistent representations from each view through projecting view-specific samples to the basis space. Specifically, we first construct the cross-view consistent basis space $\mathbf{Z}$ through multi-view samples reconstruction. As shown in Figure 2, $\mathbf{Z}$ is involved in decoding samples from multiple views and captures cross-view shared consistency, while filtering out view-specific diversity that does not overlap across the views. The decoding process of view $v$ is denoted by $D^v(\mathbf{H}^v, \mathbf{Z}) : \mathbf{H}^v \mathbf{Z} \mapsto \mathbf{X}_{pre}^v \in \mathbb{R}^{N \times D_v}$, where $\mathbf{H}^v = \{\mathbf{h}_i^v\}_{i=1}^N \in \mathbb{R}^{N \times c}$ indicates the latent representation that is extracted by encoder $E^v(\mathbf{X}^v) : \mathbf{X}^v \mapsto \mathbf{H}^v$. After that, we aim to learn a cross-view consistent structure $\mathbf{S}_{con} = \{\mathbf{s}_{coni}\}_{i=1}^c \in \mathbb{R}^{c \times c}$ to constrain the structural consistency of captured information, which further facilitates $\mathbf{Z}$ to capture consistent information. To achieve this purpose, we relax the widely used orthogonal constraint on $\mathbf{Z}$ to be linearly independent, since it is difficult for orthogonal $\mathbf{Z}$ to learn structural relationships. This relaxation is formulated as:

$$\mathbf{Z}\mathbf{Z}^T = \mathbf{I} \implies rank(\mathbf{Z}) = c, \tag{3}$$

where the similarity structure $\mathbf{S} = \{\mathbf{s}_i\}_{i=1}^c \in \mathbb{R}^{c \times c}$ of $\mathbf{Z}$ is obtained by a scaled exponential form of cosine similarity $e^{cos(\cdot,\cdot)/\tau_f}$ and is introduced to learn the consistent structure $\mathbf{S}_{con}$, where $cos(\mathbf{z}_i, \mathbf{z}_j)$ indicates the cosine similarity between $\mathbf{z}_i$ and $\mathbf{z}_j$, the exponential function is used to magnify the difference across similarity scores for obtaining a clearer similarity structure and $\tau_f$ is a temperature coefficient. However, directly introducing $\mathbf{S}$ as a structural constraint may cause $\mathbf{Z}$ to learn structurally equivalent basis vectors, inducing structural mismatch of captured information [39]. To address this issue, we design a Structure Clarity Regularization (SCR) to mitigate structural equivalence of $\{\mathbf{z}_i\}_{i=1}^c$ on $\mathbf{S}$. Considering that structurally equivalent basis vectors $(\mathbf{z}_i, \mathbf{z}_j)$ share the coincident neighbor nodes and the same structural relationships with each neighbor node $\mathbf{z}_k$ [39], i.e., $s_{i,k} = s_{j,k}$, we measure the structural equivalence between $\mathbf{z}_i$ and $\mathbf{z}_j$ as follows:

$$\varphi(\mathbf{s}_i, \mathbf{s}_j) = \sum_{\substack{k=1 \\ k \neq i,j}}^c (s_{i,k} - s_{j,k})^2, \tag{4}$$

where the higher value of $\varphi(\mathbf{s}_i, \mathbf{s}_j)$ indicates the lower structural equivalence between $\mathbf{z}_i$ and $\mathbf{z}_j$. After measuring the structural equivalence across $\{\mathbf{z}_i\}_{i=1}^c$, our Structure Clarity Regularization is designed to penalize the structural equivalence as follows:

$$\mathcal{L}_s = \sum_{1 \leqslant i < j \leqslant c} e^{-\varphi(\mathbf{s}_i, \mathbf{s}_j)/\tau_f}, \tag{5}$$

where the negative exponential function $e^{-\varphi(,)}$ encourages $\varphi(,)$ toward higher values to penalize structural equvalence across $\{\mathbf{z}_i\}_{i=1}^c$. Through the above operations, $\mathbf{Z}$ can effectively capture cross-view consistency from multiple views, promoting autoencoders to directly extract consistent representations from each view through projecting view-specific samples to the basis space. To strengthen the capability of autoencoders in extracting consistent representations, a reconstruction loss between $\{D^v(\mathbf{H}^v, \mathbf{Z})\}_{v=1}^V$ and $\{\mathbf{X}^v\}_{v=1}^V$ is introduced as follows:

$$\mathcal{L}_r = \sum_{v=1}^V \|\mathbf{X}^v - D^v(\mathbf{H}^v, \mathbf{Z})\|_F^2 = \sum_{v=1}^V \|\mathbf{X}^v - D^v(E^v(\mathbf{X}^v), \mathbf{Z})\|_F^2. \tag{6}$$

The extracted consistent representations $\{\mathbf{H}^v\}_{v=1}^V$ are used in the next stage for cross-view fusion.

## 3.3 Global contrastive fusion

To further fuse the extracted representations $\{\mathbf{H}^v\}_{v=1}^V$ while avoiding fusing non-corresponding representations that derive from different sample instances, we bypass instance-to-instance fusion and perform cross-view fusion at a global level. Specifically, we first calculate the global center $\bar{\mathbf{h}}^v$ of $\mathbf{H}^v$ in each view $v$, where $\bar{\mathbf{h}}^v = \sum_{i=1}^N \mathbf{h}_i^v / N$. After that, we select a view as central view *core* and bring $\bar{\mathbf{h}}^{core}$ closer to global centers $\{\{\bar{\mathbf{h}}^v\}_{v=1}^V, v \neq core\}$ for promoting $\mathbf{H}^{core}$ to be globally consistent with $\{\{\mathbf{H}^v\}_{v=1}^V, v \neq core\}$:

$$\mathcal{L}_c = \sum_{\substack{v=1 \\ v \neq core}}^V \left\| \bar{\mathbf{h}}^{core} - \bar{\mathbf{h}}^v \right\|_F^2, \tag{7}$$

where *core* is set to the view with the largest original feature dimension since it usually provides a more comprehensive description of samples and a more representative global center for facilitating cross-view global fusion, and $\mathbf{H}^{core}$ is treated as the fused cross-view consistent representation for subsequent clustering. However, such global-to-global operation only directly influences the global center $\bar{\mathbf{h}}^{core}$ of $\mathbf{H}^{core}$, failing to ensure that each instance $\mathbf{h}_i^{core}$ effectively fuses multi-view global information to achieve global consistency. To solve this issue, we design the Instance Global Contrastive Fusion (IGCF) to introduce instance-to-global contrast, where $\{(\mathbf{h}_i^{core}, \bar{\mathbf{h}}^v), core \neq v\}$ serves as positive pairs for bringing each instance of $\mathbf{H}^{core}$ closer to global centers $\{\{\bar{\mathbf{h}}^v\}_{v=1}^V, v \neq core\}$ while $\{(\mathbf{h}_i^{core}, \mathbf{h}_j^{core}), i \neq j\}$ serves as negative pairs for reinforcing the discriminability across $\{\mathbf{h}_i^{core}\}_{i=1}^N$. Such a strategy encourages each instance of $\mathbf{H}^{core}$ to effectively fuse multi-view global information while prompting the extracted consistent cluster structure of $\mathbf{H}^{core}$ to be clearer. In addition, we mask the normally used cross-view negative pairs $\{(\mathbf{h}_i^{core}, \mathbf{h}_j^v), core \neq v\}$, since they often include the representation pairs from the same sample in unaligned multi-view data and hinder $\{(\mathbf{H}^{core}, \mathbf{H}^v), core \neq v\}$ from achieving global consistency. Accordingly, our Instance Global Contrastive Fusion is formulated as follows:

$$\mathcal{L}_c = -\frac{1}{N} \sum_{i=1}^N \sum_{\substack{1 \leqslant v \leqslant V \\ v \neq core}} \log \frac{e^{d\left(\mathbf{h}_i^{core}, \bar{\mathbf{h}}^v\right)/\tau_l}}{\sum_{\substack{j=1 \\ j \neq i}}^N e^{d\left(\mathbf{h}_i^{core}, \mathbf{h}_j^{core}\right)/\tau_l} + N e^{d\left(\mathbf{h}_i^{core}, \bar{\mathbf{h}}^v\right)/\tau_l}}, \tag{8}$$

where $\tau_l$ is a temperature coefficient, $N e^{d\left(\mathbf{h}_i^{core}, \bar{\mathbf{h}}^v\right)/\tau_l}$ in denominator is used to prevent all instances $\{\mathbf{h}_i^{core}\}_{i=1}^N$ from collapsing onto centers to avoid poor separability.

**Theorem 1** *Assuming that* $\bar{\mathbf{H}}^v = \{\bar{\mathbf{h}}_j^v\}_{j=1}^N$, $\bar{\mathbf{h}}_j^v = \bar{\mathbf{h}}^v, j = 1, 2, \ldots, N$, *and there exists a constant* $\delta$ *such that* $p(\bar{\mathbf{h}}_i^v | \mathbf{h}_i^{core}) > \delta, i = 1, 2, \ldots, N$ *holds, then*

$$\sum_{\substack{v=1 \\ v \neq core}}^V I(\mathbf{H}^{core}, \bar{\mathbf{H}}^v) \geq (V-1) \log N - \delta \mathcal{L}_c, \tag{9}$$

Theorem 1 indicates that minimizing contrastive loss $\mathcal{L}_c$ is equal to maximizing mutual information between $\mathbf{H}^{core}$ and global centers $\{\{\bar{\mathbf{h}}^v\}_{v=1}^V, v \neq core\}$, where the detailed proof is provided in Appendix D. Finally, the fused cross-view consistent representation $\mathbf{H}^{core}$ is used for clustering with K-means. The whole loss function in our method AF-UMC is represented as:

$$\mathcal{L} = \mathcal{L}_r + \lambda \mathcal{L}_s + \gamma \mathcal{L}_c, \tag{10}$$

where $\lambda$ and $\gamma$ are trade-off coefficients.

## 3.4 Optimization

Our designed AF-UMC, consisting of multiple autoencoders and a basis matrix that indicates cross-view consistent basis space, is optimized by a gradient descent algorithm. Specifically, the autoencoders and basis matrix are trained for reconstructing original samples, where the basis matrix is optimized by Eqs. (5) and (6) for capturing structurally matched consistency, and autoencoders are optimized by Eq. (6) for extracting consistent representations from each view. Afterwards, a global contrastive fusion operation is conducted to fuse cross-view consistent representation by Eq. (8). Finally, the cross-view consistent representation is used for clustering with the K-means algorithm.

Table 1: Statistical characteristics of the ten datasets.

| Data | Samples | Clusters | View dimensions |
|---|---|---|---|
| **Caltech7-5** | 1400 | 7 | 40/254/1984/512/928 |
| **Handwritten** | 2000 | 10 | 240/76/216/47/64/6 |
| **Scene** | 4485 | 15 | 20/59/40 |
| **Caltech102-5** | 9144 | 102 | 48/40/254/512/928 |
| **Hdigit** | 10000 | 10 | 784/256 |
| **Aloi** | 10800 | 100 | 77/13/64/125 |
| **NUSWIDEOBJ** | 30000 | 31 | 65/226/145/74/129 |
| **NoisyMNIST** | 50000 | 10 | 784/784 |
| **Cifar10** | 50000 | 10 | 512/2048/1024 |
| **YoutubeFace** | 101499 | 31 | 64/512/64/647/838 |

## 4 Experiments

### 4.1 Experimental settings

**Datasets.** We employ ten widely-used multi-view datasets for comparative studies, which includes six small-scale datasets of *Caltech7-5* [4], *Handwritten* [26], *Scene* [5], *Caltech102-5* [4], *Hdigit* [1], *Aloi* [15] and four large-scale datasets *NUSWIDEOBJ* [16], *NoisyMNIST* [24], *Cifar10* [42], *YoutubeFace* [7]. The specific characteristics of these datasets are listed in Table 1.

**The compared methods.** In order to verify the effectiveness of AF-UMC, we employ six state-of-the-art unaligned multi-view clustering methods for comparative experiments on small-scale datasets, including **MVC-UM** (KDD, 2021) [40], **T-UMC** (TCYB, 2022) [18], **UPMGC** (TNNLS, 2023) [32], **FUMC** (IJCAI, 2024) [14], **OpVuC** (TMM, 2024) [3], **TUMCR** (KDD, 2024) [8]. Besides, considering that most unaligned multi-view clustering methods cannot be employed on large-scale datasets due to its excessive complexity, except for **FUMC** and **OpVuC**, we additionally employ four state-of-the-art aligned multi-view clustering methods for unaligned large-scale datasets, including **LMVSC** (AAAI, 2020) [10], **MFLVC** (CVPR, 2022) [37], **GCFAgg** (CVPR, 2023) [38] and **SCMVC** (TMM, 2024) [33]. Moreover, for the reliability of our comparative experiments, all compared methods are implemented according to the source codes released by the authors, and the optimal parameters are set according to the suggestions in the corresponding literature.

**Evaluation metrics.** There are four widely-used metrics applied to quantitatively evaluate the performance of unaligned multi-view clustering methods, including Accuracy (ACC), Normalized Mutual Information (NMI), Purity (Pur) and Adjusted Rand Index (ARI), whose detailed definitions are illustrated in [17]. For each of the above metrics, the higher value indicates the better performance.

**Implementation details.** The encoder $E^v$ and decoder $D^v$ are respectively formulated by MLPs with dimensions $\{D_v, 500, 500, 2000, 512, c\}$ and $\{d, 2000, 500, 500, D_v\}$, where the activation function is ReLU. The consistent basis space $\mathbf{Z}$ is set to a matrix of $c \times d$, where $c$ is set to the number of categories $k$ and $d$ is set to 512. During the whole process, AF-UMC trains 50 epochs on mini-batches of size 256 by using Adam optimizer [11] with a learning rate of 0.0003 in PyTorch [23] framework. The hyperparameters $\gamma$ and $\lambda$ are set to 1 and 1, respectively. All experiments are conducted on the same machine with the Intel(R) Xeon(R) Gold 6148 2.40GHz CPU, 8 GeForce RTX 3090 GPUs, and 512GB RAM.

### 4.2 Experimental results

Table 2 and Table 3 respectively record the experimental comparisons on small-scale datasets and large-scale datasets, where the best and the second-best performance are highlighted in bold and underlined, respectively. In addition, Figure 3 illustrates the visualization of clustering results of each method on the *Handwritten* dataset. According to Tables 2-3 and Figure 3, we can observe that:

(1) In Tables 2-3, except for ARI on *Scene* dataset, our AF-UMC is superior to all comparing methods on all evaluation metrics, even has a significant leading gap compared with second-best

Table 2: Comparative results between AF-UMC and 6 state-of-the-art methods on six small-scale datasets, where the best results are presented in **bold** and the second-best are in underline.

| Dataset | Metric | Method | | | | | | |
|---------|--------|--------|-------|------|------|-------|-------|--------|
| | | MVC-UM | UPMGC | FUMC | OpVuC | TUMCR | T-UMC | AF-UMC |
| **Caltech7-5** | ACC | 0.2785 | 0.8079 | 0.2044 | 0.3279 | 0.2557 | 0.4079 | **0.8721** |
| | NMI | 0.1038 | 0.7137 | 0.0269 | 0.1229 | 0.0721 | 0.3271 | **0.7798** |
| | ARI | 0.0875 | 0.7034 | 0.0137 | 0.0820 | 0.0612 | 0.3180 | **0.7485** |
| | PUR | 0.3014 | 0.8079 | 0.2098 | 0.3851 | 0.2771 | 0.2771 | **0.8721** |
| **Handwritten** | ACC | 0.7465 | 0.6270 | 0.1946 | 0.1465 | 0.4830 | 0.7720 | **0.9035** |
| | NMI | 0.7230 | 0.5860 | 0.0652 | 0.0181 | 0.3853 | 0.6703 | **0.8205** |
| | ARI | 0.6305 | 0.5014 | 0.0431 | 0.0052 | 0.3627 | 0.6564 | **0.8002** |
| | PUR | 0.7465 | 0.6346 | 0.2004 | 0.1705 | 0.4980 | 0.7720 | **0.9035** |
| **Scene** | ACC | 0.2608 | 0.1386 | 0.1647 | 0.3275 | 0.2990 | 0.3882 | **0.4190** |
| | NMI | 0.2787 | 0.0576 | 0.0904 | 0.3409 | 0.2488 | 0.3816 | **0.4217** |
| | ARI | 0.1947 | 0.0434 | 0.0731 | 0.1836 | 0.2171 | **0.2856** | 0.2548 |
| | PUR | 0.2791 | 0.1503 | 0.1727 | 0.3741 | 0.3398 | 0.4239 | **0.4593** |
| **Caltech102-5** | ACC | 0.0638 | 0.0930 | 0.0576 | 0.1355 | 0.0977 | 0.1017 | **0.2275** |
| | NMI | 0.2150 | 0.1883 | 0.1472 | 0.2974 | 0.2066 | 0.2512 | **0.4528** |
| | ARI | 0.0571 | 0.0742 | 0.0431 | 0.0958 | 0.0639 | 0.0741 | **0.1755** |
| | PUR | 0.1846 | 0.1704 | 0.1226 | 0.2794 | 0.1943 | 0.2348 | **0.4269** |
| **Hdigit** | ACC | 0.4627 | 0.4087 | 0.3603 | 0.3994 | 0.1551 | 0.4993 | **0.6950** |
| | NMI | 0.4418 | 0.3700 | 0.3355 | 0.3151 | 0.0255 | 0.4387 | **0.6000** |
| | ARI | 0.3987 | 0.2943 | 0.0216 | 0.2054 | 0.0203 | 0.3708 | **0.5194** |
| | PUR | 0.5031 | 0.4545 | 0.4289 | 0.4173 | 0.1679 | 0.5398 | **0.6980** |
| **Aloi** | ACC | 0.3543 | 0.0383 | 0.0874 | 0.1432 | 0.2330 | 0.5057 | **0.5399** |
| | NMI | 0.6039 | 0.1118 | 0.2220 | 0.4105 | 0.3363 | 0.6536 | **0.7590** |
| | ARI | 0.2381 | 0.0213 | 0.0351 | 0.0897 | 0.1853 | 0.3859 | **0.4151** |
| | PUR | 0.3692 | 0.0402 | 0.0897 | 0.1747 | 0.2429 | 0.5238 | **0.5816** |

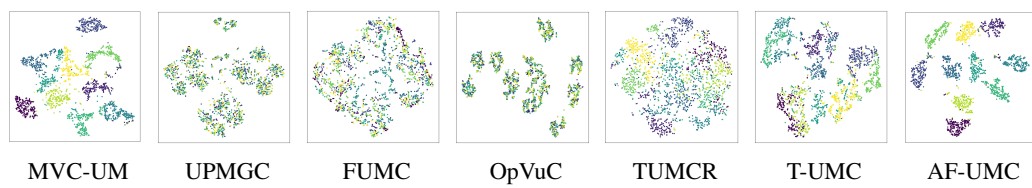

MVC-UM    UPMGC    FUMC    OpVuC    TUMCR    T-UMC    AF-UMC

Figure 3: The visualizations of the clustering results of different methods on *Handwritten* dataset.

methods. Especially on the *Hdigit* dataset, the improvements over the second-best method are 19.57%, 15.57%, 10.99%, and 15.82% on ACC, NMI, ARI and PUR, respectively. These experimental results demonstrate the effectiveness of AF-UMC and we attribute such success to our designed alignment-free consistency fusion framework, which bypasses undesired alignment strategies and obtains a cross-view consistent representation with a clearer cluster structure through global fusion.

(2) In Figure 3, we select all unaligned multi-view clustering methods to conduct the visualization comparisons of clustering results with our proposed AF-UMC. We can observe that our AF-UMC exhibits a clearer cluster structure than all other methods, which demonstrates the superiority of AF-UMC in fusing consistent representation from unaligned multi-view data.

## 4.3 Model analysis

**Convergence analysis.** Figure 4 shows the convergence curves of AF-UMC on *Caltech7-5*, *NoisyM-NIST* datasets, where the values of loss and evaluation metrics are illustrated in each subfigure. According to Figure 4, we can observe that the loss drops significantly at the beginning of the iteration process and then gradually reaches a stable value as the number of iterations increases. And the evaluation metrics gradually increase and fluctuate in a narrow range. These results verify the convergence of our proposed AF-UMC.

Table 3: Comparative results between AF-UMC and 6 state-of-the-art methods on four large-scale datasets. "-" means that the code can't be run due to its excessive time or space complexity.

| Dataset | Metric | Method | | | | | | |
|---|---|---|---|---|---|---|---|---|
| | | LMVSC | MFLVC | GCFAgg | SCMVC | FUMC | OpVuC | AF-UMC |
| **NUSWIDEOBJ** | ACC | 0.0674 | 0.0973 | 0.0455 | 0.0474 | 0.0945 | 0.1016 | **0.1216** |
| | NMI | 0.0263 | 0.0047 | 0.0057 | 0.0089 | 0.0775 | 0.0864 | **0.1041** |
| | ARI | 0.0158 | 0.0004 | 0.0002 | 0.0008 | 0.0193 | 0.0213 | **0.0306** |
| | PUR | 0.0842 | 0.1268 | 0.1223 | 0.1269 | 0.1914 | 0.2041 | **0.2208** |
| **NoiyMNIST** | ACC | 0.2416 | 0.1131 | 0.1078 | 0.1311 | 0.2843 | 0.5111 | **0.5899** |
| | NMI | 0.1512 | 0.0015 | 0.0006 | 0.0099 | 0.2329 | 0.4241 | **0.4982** |
| | ARI | 0.1835 | 0.0007 | 0.0001 | 0.0051 | 0.2231 | 0.3308 | **0.4154** |
| | PUR | 0.2908 | 0.1137 | 0.1088 | 0.1357 | 0.3457 | 0.5377 | **0.6247** |
| **Cifar10** | ACC | 0.3961 | 0.3550 | 0.1284 | 0.3831 | 0.2174 | 0.8008 | **0.8453** |
| | NMI | 0.3323 | 0.1779 | 0.0067 | 0.1975 | 0.0892 | 0.6872 | **0.7025** |
| | ARI | 0.3178 | 0.1074 | 0.0033 | 0.1464 | 0.0747 | 0.6284 | **0.6932** |
| | PUR | 0.4966 | 0.3552 | 0.1324 | 0.3969 | 0.2191 | 0.8008 | **0.8453** |
| **YoutubeFace** | ACC | 0.0405 | 0.0737 | 0.0414 | 0.0510 | 0.0717 | - | **0.1625** |
| | NMI | 0.0169 | 0.0049 | 0.0029 | 0.0187 | 0.0366 | - | **0.1444** |
| | ARI | 0.0105 | 0.0008 | 0.0001 | 0.0018 | 0.0068 | - | **0.0270** |
| | PUR | 0.1132 | 0.2662 | 0.2662 | 0.2662 | 0.2662 | - | **0.2851** |

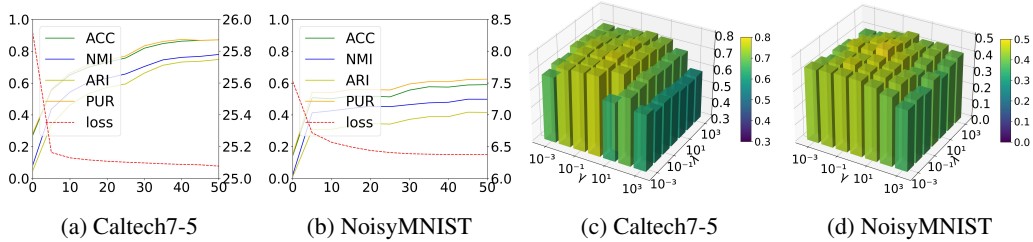

(a) Caltech7-5          (b) NoisyMNIST          (c) Caltech7-5          (d) NoisyMNIST

Figure 4: The convergence analysis and parameter analysis on *Caltech7-5* and *NoisyMNIST* datasets.

**Parameter sensitivity analysis.**    We experimentally evaluate the effect of hyperparameters on the clustering performance of AF-UMC, which includes $\gamma$ and $\lambda$. Figure 4 shows the NMI metric value of AF-UMC on *Caltech7-5*, *NoisyMNIST* datasets, where $\gamma$ and $\lambda$ are varied from $10^{-3}$ to $10^3$. According to Figure 4, the clustering results of AF-UMC are insensitive to both $\gamma$ and $\lambda$ ranging from 0.1 to 10. In our experiments, we set $\gamma$ and $\lambda$ to 1.

**Ablation study.**    We conduct two series of ablation studies from the perspective of loss functions and model components on *Caltech7-5* and *NoisyMNIST* datasets. Table 4 records the ablation studies of different loss functions, where $\mathcal{L}_r$ is the loss to reconstruct original samples, $\mathcal{L}_s$ is the loss to capture structurally matched consistency and $\mathcal{L}_c$ is the loss to globally fuse the extracted representations. Table 5 records the ablation studies of different model components, where ***BAE*** represents the autoencoders with consistent basis space and ***Ins-Glo*** represents the instance-to-global contrast operation. According to Tables 4-5, we can find that:

(1) In Table 4, (C) is superior to (B), which indicates that capturing structurally matched consistency into basis space is helpful in autoencoders extracting consistent representations from each view and further improves the performance of cross-view fusion. Meanwhile, (C) also shows better clustering performance than (A), which indicates that our globally fused cross-view consistent representation contains a clearer cluster structure for achieving better clustering performance.

(2) In Table 5, (a) replaces the designed instance-to-global contrast operation with global-to-global operation as Eq. (7), and (b) replaces the ***BAE*** with traditional autoencoders. According to Table 5, (c) shows better performance than (a), which indicates that our designed instance-to-global contrast operation effectively fuses multi-view samples into a cross-view consistent representation for more effective clustering. Meanwhile, (c) outperforms (b), which demonstrates that the ***BAE*** successfully extracts consistent representations from each view for promoting subsequent global cross-view fusion.

Table 4: Ablation studies on loss functions of AF-UMC on *Caltech7-5* and *NoisyMNIST* datasets.

| | Loss | | | Caltech7-5 | | | | NoisyMNIST | | | |
|---|---|---|---|---|---|---|---|---|---|---|---|
| | $\mathcal{L}_r$ | $\mathcal{L}_s$ | $\mathcal{L}_c$ | ACC | NMI | PUR | ARI | ACC | NMI | PUR | ARI |
| (A) | ✓ | ✓ | | 0.8079 | 0.7422 | 0.8164 | 0.6880 | 0.4860 | 0.4063 | 0.5257 | 0.2835 |
| (B) | ✓ | | ✓ | 0.8014 | 0.6983 | 0.8014 | 0.6406 | 0.5046 | 0.4573 | 0.5501 | 0.3297 |
| (C) | ✓ | ✓ | ✓ | **0.8721** | **0.7798** | **0.8721** | **0.7485** | **0.5899** | **0.4982** | **0.6247** | **0.4154** |

Table 5: Ablation studies on model components of AF-UMC on *Caltech7-5* and *NoisyMNIST* datasets.

| | Components | | Caltech7-5 | | | | NoisyMNIST | | | |
|---|---|---|---|---|---|---|---|---|---|---|
| | *BAE* | *Ins-Glo* | ACC | NMI | PUR | ARI | ACC | NMI | PUR | ARI |
| (a) | ✓ | | 0.8107 | 0.7498 | 0.8107 | 0.6820 | 0.4826 | 0.4256 | 0.5328 | 0.2919 |
| (b) | | ✓ | 0.7107 | 0.5989 | 0.7107 | 0.5312 | 0.4793 | 0.4136 | 0.5202 | 0.2862 |
| (c) | ✓ | ✓ | **0.8721** | **0.7798** | **0.8721** | **0.7485** | **0.5899** | **0.4982** | **0.6247** | **0.4154** |

## 5    Conclusion

In this paper, we propose an alignment-free consistency fusion framework named AF-UMC for unaligned multi-view clustering. Different from previous methods that conduct view-alignment then fuse aligned feature representations, our proposed method directly extracts consistent representations from each view for global multi-view fusion. Our proposed method significantly mitigates the degraded performance caused by undesired view-alignment results in previous methods while greatly reducing algorithm complexity and enhancing its efficiency. Extensive experimental results on various datasets have verified the effectiveness of our proposed method.

## 6    Acknowledgments

This work was supported by the National Natural Science Foundation of China (No. 62306020), the Young Elite Scientist Sponsorship Program by BAST (No. BYESS2024199), the Beijing Natural Science Foundation (No. L244009), the National Key Research and Development Program of China (No. 2023YFB3107100), and the Central Guidance for Local Scientific and Technological Development Fund (No.2024ZY0124).

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

# Appendix

We provide more details and results about our work in the appendices. Here are the contents:

- Appendix A: Commonly used notations.
- Appendix B: Training process of AF-UMC.
- Appendix C: Complexity analysis of AF-UMC.
- Appendix D: Proof of Theorem 1.
- Appendix E: Additional experiment results.

## A  Commonly used notations

Table 6 shows the commonly used notations and the associated definitions.

Table 6: Notations and Definitions

| Notation | Definition |
|---|---|
| $\mathbf{X}^v$ | The samples of the $v$-th view |
| $\mathbf{Z}^v$ | The basis space of the $v$-th view |
| $\mathbf{Z}$ | The cross-view consistent basis space |
| $\{\mathbf{z}_i\}_{i=1}^c$ | The $c$ basis vectors in $\mathbf{Z}$ |
| $E^v(\cdot)$ | The encoder of the $v$-th view |
| $D^v(\cdot)$ | The decoder of the $v$-th view |
| $\mathbf{H}^v$ | The extracted representation of the $v$-th view |
| $\mathbf{S}$ | The similarity structure across basis vectors $\{\mathbf{z}_i\}_{i=1}^c$ |
| $\bar{\mathbf{h}}^v$ | The global center of $\mathbf{H}^v$ |

## B  Training process of AF-UMC

Algorithm 1 outlines the execution flow for AF-UMC. At each training epoch $t$, autoencoders first extract consistent representations $\{\mathbf{H}^v\}_{v=1}^V$ by projecting multi-view samples $\{\mathbf{X}^v\}_{v=1}^V$ onto the cross-view consistent basis space $\mathbf{Z}$. Then, global contrastive fusion globally pulled together these extracted representations $\{\mathbf{H}^v\}_{v=1}^V$ to fuse a cross-view consistent representation $\mathbf{H}^{core}$. After $T$ training epochs, the final clustering results are obtained by performing K-means clustering on $\mathbf{H}^{core}$.

---

**Algorithm 1** The Training Process of AF-UMC.

---

**Input:** Unaligned multi-view data $\mathbf{X} = \{\mathbf{X}^v\}_{v=1}^V$, number of clusters $c$, training epochs $T$.
**Output:** Clustering results
 1: Initialize autoencoders $\{E^v(\cdot), D^v(\cdot)\}_{v=1}^V$ and cross-view consistent basis space $\mathbf{Z}$.
 2: **for** epoch $t = 1$ to $T$:
 3:  **for** view $v = 1$ to $V$:
 4:   Extract consistent representation $\mathbf{H}^v$ by projecting $\mathbf{X}^v$ onto $\mathbf{Z}$.
 5:  **end for**
 6:  Globally bring $\{\mathbf{H}^v\}_{v=1}^V$ closer to fuse a cross-view consistent representation $\mathbf{H}^{core}$.
 7:  Optimize model by $\mathcal{L}_r$, $\mathcal{L}_s$ and $\mathcal{L}_c$.
 8: **end for**
 9: Perform K-means clustering on $\mathbf{H}^{core}$.

---

## C  Complexity analysis of AF-UMC

We analyze our proposed AF-UMC in terms of space/time complexity.

**Space Complexity:** In our method, the memory costs contain a basis space matrix $\mathbf{Z} \in \mathbb{R}^{c \times d}$, $V$ autoencoders and $V$ representation matrices $\{\mathbf{H}^v\}_{v=1}^V \in \mathbb{R}^{N \times c}$, where the space complexity of an

autoencoder is $\mathcal{O}(lNd)$ and $l$ is the number of MLP layers. As a result, the total space complexity of our AF-UMC is $\mathcal{O}(cd + VlNd + VNc)$.

**Time Complexity:** The time cost of AF-UMC arises from three parts: (1) $\mathcal{O}(VNd + c^3 + NV^2d)$, the cost of computing three loss functions. (2) $\mathcal{O}(VlNd)$, the cost of optimizing $V$ autoencoders. (3) $\mathcal{O}(cd)$, the cost of optimizing a cross-view consistent basis space $\mathbf{Z}$. Therefore, the total time cost of AF-UMC is $\mathcal{O}(VNd + c^3 + NV^2d + VlNd + cd)$.

## D   Proof of theorem 1

In this part, we want to prove that minimizing contrastive loss $\mathcal{L}_c$ is equal to maximizing mutual information. For expressing more clearly, we first construct $\bar{\mathbf{H}}^v = \{\bar{\mathbf{h}}_j^v\}_{j=1}^N$, where $\bar{\mathbf{H}}^v \in \mathbb{R}^{N \times c}$ and $\bar{\mathbf{h}}_j^v = \bar{\mathbf{h}}^v$ indicates the $j$-th row of $\bar{\mathbf{H}}^v$. The proof is motivated by [22, 43].

*Proof.* $\mathcal{L}_c$ is our designed contrastive loss, which is formulated as:

$$\mathcal{L}_c = -\frac{1}{N}\sum_{i=1}^N \sum_{\substack{1 \leqslant v \leqslant V \\ v \neq core}} \log \frac{e^{d\left(\mathbf{h}_i^{core}, \bar{\mathbf{h}}^v\right)/\tau_l}}{\sum_{\substack{i,j=1 \\ i \neq j}}^N e^{d\left(\mathbf{h}_i^{core}, \mathbf{h}_j^{core}\right)/\tau_l} + N e^{d\left(\mathbf{h}_i^{core}, \bar{\mathbf{h}}^v\right)/\tau_l}},$$

We assume that $p(\mathbf{h}_i^{core}, \bar{\mathbf{h}}_j^v) = p(\mathbf{h}_i^{core})p(\bar{\mathbf{h}}_j^v), i \neq j$ and let $\mathcal{N}_i = \sum_{j=1}^N \frac{p(\mathbf{h}_i^{core}, \bar{\mathbf{h}}_j^v)}{p(\mathbf{h}_i^{core})p(\bar{\mathbf{h}}_j^v)}$, we have:

$$
\begin{aligned}
I\left(\mathbf{H}^{core}; \bar{\mathbf{H}}^v\right) &= \sum_{i=1}^N \sum_{j=1}^N p\left(\mathbf{h}_i^{core}, \bar{\mathbf{h}}_j^v\right) \log \frac{p\left(\mathbf{h}_i^{core}, \bar{\mathbf{h}}_j^v\right)}{p\left(\mathbf{h}_i^{core}\right) p\left(\bar{\mathbf{h}}_j^v\right)} \\
&= \sum_{i=1}^N p\left(\mathbf{h}_i^{core}, \bar{\mathbf{h}}_i^v\right) \log \frac{p\left(\mathbf{h}_i^{core}, \bar{\mathbf{h}}_i^v\right)}{p\left(\mathbf{h}_i^{core}\right) p\left(\bar{\mathbf{h}}_i^v\right)} + \sum_{i=1}^N \sum_{j \neq i} p\left(\mathbf{h}_i^{core}, \bar{\mathbf{h}}_j^v\right) \log \frac{p\left(\mathbf{h}_i^{core}, \bar{\mathbf{h}}_j^v\right)}{p\left(\mathbf{h}_i^{core}\right) p\left(\bar{\mathbf{h}}_j^v\right)} \\
&= \sum_{i=1}^N p\left(\mathbf{h}_i^{core}, \bar{\mathbf{h}}_i^v\right) \log \left(\frac{p\left(\mathbf{h}_i^{core}, \bar{\mathbf{h}}_i^v\right)}{p\left(\mathbf{h}_i^{core}\right) p\left(\bar{\mathbf{h}}_i^v\right) \cdot \mathcal{N}_i} \cdot \mathcal{N}_i\right) \\
&= \sum_{i=1}^N p\left(\mathbf{h}_i^{core}, \bar{\mathbf{h}}_i^v\right) \log \frac{\frac{p(\mathbf{h}_i^{core}, \bar{\mathbf{h}}_i^v)}{p(\mathbf{h}_i^{core})p(\bar{\mathbf{h}}_i^v)}}{\mathcal{N}_i} + \sum_{i=1}^N p\left(\mathbf{h}_i^{core}, \bar{\mathbf{h}}_i^v\right) \log \mathcal{N}_i.
\end{aligned}
$$

Since positive pairs are correlated, we have the estimate: $p(\mathbf{h}_i^{core}, \bar{\mathbf{h}}_i^v) \geq p(\mathbf{h}_i^{core})p(\bar{\mathbf{h}}_i^v)$. According to [36], we have $p(\mathbf{h}_i^{core}) \approx \frac{1}{N}, i = 1, 2, \ldots, N$, and $e^{d\left(\mathbf{h}_i^{core}, \bar{\mathbf{h}}_j^v\right)/\tau_g} \propto \frac{p(\mathbf{h}_i^{core}, \bar{\mathbf{h}}_j^v)}{p(\mathbf{h}_i^{core})p(\bar{\mathbf{h}}_j^v)}$, then:

$$
\begin{aligned}
\sum_{\substack{v=1 \\ v \neq core}}^V I(\mathbf{H}^{core}, \bar{\mathbf{H}}^v) &= \sum_{v=1}^V \sum_{i=1}^N p\left(\mathbf{h}_i^{core}, \bar{\mathbf{h}}_i^v\right) \log \frac{\frac{p(\mathbf{h}_i^{core}, \bar{\mathbf{h}}_i^v)}{p(\mathbf{h}_i^{core})p(\bar{\mathbf{h}}_i^v)}}{\mathcal{N}_i} \\
&+ \sum_{\substack{v=1 \\ v \neq core}}^V \sum_{i=1}^N p\left(\mathbf{h}_i^{core}, \bar{\mathbf{h}}_i^v\right) \log \left(\sum_{j=1}^N \frac{p\left(\mathbf{h}_i^{core}, \bar{\mathbf{h}}_j^v\right)}{p\left(\mathbf{h}_i^{core}\right) p\left(\bar{\mathbf{h}}_j^v\right)}\right) \\
&\approx \sum_{\substack{v=1 \\ v \neq core}}^V \sum_{i=1}^N \frac{1}{N} p\left(\bar{\mathbf{h}}_i^v \mid \mathbf{h}_i^{core}\right) \log \frac{\frac{p(\mathbf{h}_i^{core}, \bar{\mathbf{h}}_i^v)}{p(\mathbf{h}_i^{core})p(\bar{\mathbf{h}}_i^v)}}{\mathcal{N}_i} \\
&+ \sum_{\substack{v=1 \\ v \neq core}}^V \log \left(N - 1 + \frac{p\left(\mathbf{h}_i^{core}, \bar{\mathbf{h}}_i^v\right)}{p\left(\mathbf{h}_i^{core}\right) p\left(\bar{\mathbf{h}}_i^v\right)}\right) \\
&\geq \frac{\delta}{N} \sum_{\substack{v=1 \\ v \neq core}}^V \sum_{i=1}^N \log \frac{e^{sim(\mathbf{h}_i^{core}, \bar{\mathbf{h}}_i^v)/\tau_l}}{\sum_{j \neq i} e^{\sin\left(\mathbf{h}_i^{core}, \bar{\mathbf{h}}_j^v\right)/\tau_l} + e^{sim\left(\mathbf{h}_i^{core}, \bar{\mathbf{h}}_i^v\right)/\tau_l}} + (V-1)\log N \\
&\geq (V-1)\log N - \delta\mathcal{L}_c.
\end{aligned}
$$

# E   Additional experiment results

**Visual comparison on large-scale dataset.**   Figure 5 shows the visual comparison between our AF-UMC and the existing SOTA methods (**LMVSC** [10], **MFLVC** [37], **GCFAgg** [38], **SCMVC** [33], **FUMC** [14], **OpVuC** [3]) on the large-scale dataset *Cifar10*. We can observe that our AF-UMC exhibits a clearer cluster structure than all other methods, which demonstrates the superiority of AF-UMC in fusing large-scale unaligned multi-view data.

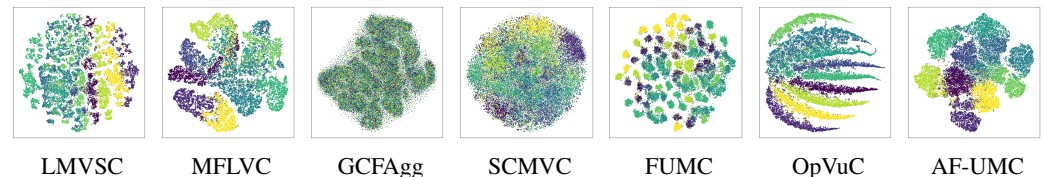

| LMVSC | MFLVC | GCFAgg | SCMVC | FUMC | OpVuC | AF-UMC |

Figure 5: The visualizations of the clustering results of different methods on *Cifar10* dataset.

**Additional Ablation study.**   **(1) Ablation study on reconstruction loss $\mathcal{L}_r$:** Table 7 shows the ablation study on $\mathcal{L}_r$. From Table 7, (B) shows a significant performance improvement over (A), indicating that $\mathcal{L}_r$ plays a critical role in improving representation quality. **(2) Ablation study on cross-view consistent basis space Z:** Considering that ablating **Z** also removes the loss $\mathcal{L}_s$ defined on **Z**, it is difficult to directly evaluate the impact of the individual basis space **Z**. To address this issue, we design the following ablation study in Table 8, where (a) ablates both $\mathcal{L}_s$ and **Z**, and (b) only ablates $\mathcal{L}_s$. From Table 8, (b) shows better performance than (a), demonstrating the effectiveness of cross-view consistent basis space in prompting autoencoders to extract consistent representations from each view.

Table 7: Ablation studies on loss functions of AF-UMC on *Caltech7-5* and *NoisyMNIST* datasets.

|     | Loss | Caltech7-5 | | | | NoisyMNIST | | | |
| --- | --- | --- | --- | --- | --- | --- | --- | --- | --- |
|     | $\mathcal{L}_r$ | ACC | NMI | PUR | ARI | ACC | NMI | PUR | ARI |
| (A) |     | 0.5014 | 0.4366 | 0.5429 | 0.3056 | 0.4637 | 0.3924 | 0.5057 | 0.2826 |
| (B) | ✓ | **0.8721** | **0.7798** | **0.8721** | **0.7485** | **0.5899** | **0.4982** | **0.6247** | **0.4154** |

Table 8: Ablation studies on model components of AF-UMC on *Caltech7-5* and *NoisyMNIST* datasets.

|     | Components | Caltech7-5 | | | | NoisyMNIST | | | |
| --- | --- | --- | --- | --- | --- | --- | --- | --- | --- |
|     | Z | ACC | NMI | PUR | ARI | ACC | NMI | PUR | ARI |
| (a) | w/o ($\mathcal{L}_s$ & **Z**) | 0.7107 | 0.5989 | 0.7107 | 0.5312 | 0.4793 | 0.4136 | 0.5202 | 0.2862 |
| (b) | w/o $\mathcal{L}_s$ | **0.8014** | **0.6983** | **0.8014** | **0.6406** | **0.5046** | **0.4573** | **0.5501** | **0.3297** |

