# OpenReview forum: "AF-UMC: An Alignment-Free Fusion Framework for Unaligned Multi-View Clustering"
_NeurIPS.cc/2025/Conference — NeurIPS 2025 poster_

### Official Review · Reviewer_SXo8 · 2025-06-20

**Clarity:** 3
**Significance:** 2
**Originality:** 3
**Rating:** 5
**Confidence:** 5

**Summary:**

For the task of unaligned multi-view clustering (UMC), the authors present a new alignment-free consistency fusion framework (AF-UMC). Unlike traditional methods that employ various alignment strategies to obtain aligned sample representations for cross-view fusion, AF-UMC is designed to directly extract a consistent representation from each view for global cross-view contrastive fusion. Through the operation of alignment-free consistency extraction and global contrastive fusion, AF-UMC obtains a consistent representation with a clearer clustering structure. Experimental results also demonstrate the superior performance of AF-UMC.

**Questions:**

Q1. The authors constrain the cross-view consistent basis space to reconstruct samples of multiple views in Section 3.2, and they state to use this operation for capturing the cross-view consistency and filtering out the view-specific diversity. Why is such an operation able to encourage the basis space to capture consistency while filtering out diversity?
Q2. Why do the authors implement a cross-view consistent basis space? What are the advantages of the cross-view consistent basis space in comparison to current methods that respectively construct a basis space in each view?
Q3. Why do the authors set the limitation that the dimension $d$ of the basis vectors is greater than the number $c$ of basis vectors in Eq. (3)? Are there any negative effects if $c > d$?

**Ethical Concerns:**

["NO or VERY MINOR ethics concerns only"]

**Final Justification:**

All my concerns has been addressed and I decide to improve my ratings to 5.

**Limitations:**

None.

**Quality:**

3

**Strengths And Weaknesses:**

Strengths:
S1. Interesting model design: AF-UMC bypasses the traditional view-alignment operation and directly extracts a consistent representation from each view for global fusion rather than traditional alignment-based fusion.

S2. Comprehensive experimental settings: various experimental results on benchmark datasets are provided to demonstrate the effectiveness of AF-UMC, particularly the ablation study highlighting the contribution of different loss functions and model components for clustering performance in Tables 4-5.


Weaknesses:
W1. Some mathematical symbols are explained repeatedly in both Introduction and Method, which seem redundant. The authors can introduce these symbols in a single subsection or directly provide a symbol list to help readers understand.
W2. The authors divide their proposed AF-UMC into two stages: alignment-free consistency extraction and global contrastive fusion, but do not clarify how the two stages are trained, such as firstly extracting consistent representations until the reconstruction loss converges and then realizing global contrastive fusion, or fulfilling consistent representation extraction and global fusion at each training epoch. The authors should provide the complete process of model training.
W3. The authors note that AF-UMC saves the additional cost for the alignment strategy, but the paper doesn’t provide the model complexity analysis.

---

> ### Author Rebuttal · Authors · 2025-07-29
>
> ************ **For Weakness #1:** ************
> --------------------------------------------------------------------
>
> We provide a symbol list in the following table and we will add it in the final version.
>
> |    **Notation**     |  **Definition**    |
> |----------------|----------------|
> |   $\mathbf{X}^v$    | The samples of $v$-th view  |
> |  $\mathbf{Z}^v$  |    The basis space of $v$-th view  |
> |   $\mathbf{B}$  |  The cross-view consistent basis space  |
> |   $\\{\mathbf{b}\_i\\}\_{i=1}^c$ |  The $c$ basis vectors in $\mathbf{B}$|
> | $E^v(\cdot)$ | The encoder of $v$-th view |
> | $D^v(\cdot)$ | The decoder of $v$-th view |
> |   $\mathbf{H}^v$  |  The extracted representation of $v$-th view |
> |   $\mathbf{S}$  |  The similarity structure across basis vectors $\\{\mathbf{b}_i\\}\_{i=1}^c$ |
> | $\bar{\mathbf{h}}^v$ | The global center of $\mathbf{H}^v$ |
>
> --------------------------------------------------------------------
>
> ************ **For Weakness #2:** ************
> --------------------------------------------------------------------
>
> We provide the detailed training process of our proposed AF-UMC in Algorithm 1.
>
>
> --------------------------------------------------------------------
>
> **Algorithm 1: Training process of AF-UMC**
>
> **Input:** Unaligned multi-view data $\mathbf{X} = \left\\{\mathbf{X}^v\right\\}\_{v=1}^V$, number of clusters $c$, number of training epochs $T$.
>
> **Output:** Clustering results.
>
> a. Initialize autoencoders $\\{E^{v}(\cdot), D^v(\cdot)\\}\_{v=1}^V$ and cross-view consistent basis space $\mathbf{B}$.
>
> b. **for** epoch $t = 1$ to $T$:
>
> &nbsp;&nbsp;&nbsp;&nbsp;&nbsp;&nbsp;&nbsp;**for** view $v = 1$ to $V$:
>
> &nbsp;&nbsp;&nbsp;&nbsp;&nbsp;&nbsp;&nbsp;&nbsp;&nbsp;&nbsp;&nbsp;Extract consistent representation $\mathbf{H}^v$ by projecting $\mathbf{X}^v$ into $\mathbf{B}$
>
> &nbsp;&nbsp;&nbsp;&nbsp;&nbsp;&nbsp;&nbsp;**end for**
>
> &nbsp;&nbsp;&nbsp;&nbsp;&nbsp;&nbsp;&nbsp;Globally bring $\\{\mathbf{H}^v\\}\_{v=1}^V$ closer to fuse a cross-view consistent representation $\mathbf{H}^{core}$.
>
> &nbsp;&nbsp;&nbsp;&nbsp;&nbsp;&nbsp;&nbsp;Optimize model by $\mathcal{L}\_{r}$,  $\mathcal{L}\_{s}$ and $\mathcal{L}\_{c}$.
>
> &nbsp;&nbsp;&nbsp;**end for**
>
> c. Perform k-means clustering on $\mathbf{H}^{core}$.
>
> --------------------------------------------------------------------
>
> ************ **For Weakness #3:** ************
> --------------------------------------------------------------------
>
> We analyze our proposed AF-UMC in terms of space/time complexity.
>
> **Space Complexity:** In our method, the memory costs contain a basis space matrix $\mathbf{B} \in \mathbb{R}^{c\times d}$, $V$ autoencoders and $V$ representation matrices $\\{\mathbf{H}^v\\}\_{v=1}^V \in \mathbb{R}^{n\times c}$, where the space complexity of an autoencoder is $\mathcal{O}(lnd)$ and $l$ is the number of MLP layers. As a result, the total space complexity of our AF-UMC is $\mathcal{O}(cd+Vlnd+Vnc)$.
>
> **Time Complexity:** The time cost of AF-UMC arises from three parts: (1) $\mathcal{O}(Vnd+c^3+nV^2d)$, the cost of computing three loss functions. (2) $\mathcal{O}(Vlnd)$, the cost of optimizing $V$ autoencoders. (3) $\mathcal{O}(cd)$, the cost of optimizing a cross-view consistent basis space $\mathbf{B}$. Therefore, the total time cost of AF-UMC is $\mathcal{O}(Vnd+c^3+nV^2d+Vlnd+cd)$.
>
> --------------------------------------------------------------------
>
> ************ **For Question #1:** ************
> --------------------------------------------------------------------
>
> Since the basis space is shared across multiple views and optimized to minimize the overall reconstruction loss, components that are consistently present in all views contribute more effectively to reducing the overall loss. Therefore, when the basis space is designed to reconstruct samples from multiple views, it naturally prioritizes capturing cross-view consistency while gradually reducing the diversity with the model training, thereby capturing consistency while filtering out diversity.
>
> --------------------------------------------------------------------
>
> ************ **For Question #2:** ************
> --------------------------------------------------------------------
>
>
>
> Compared with existing methods that construct a basis space to capture intra-view information in each view, our cross-view consistent basis space can directly capture cross-view consistent information, facilitating autoencoders to directly extract consistent representations from each view for promoting subsequent global contrastive fusion across views.
>
> --------------------------------------------------------------------
>
> ************ **For Question #3:** ************
> --------------------------------------------------------------------
>
>
> If the number $c$ of basis vectors is greater than their dimension $d$, at least one basis vector $\mathbf{b}_i$ can be linearly represented by the others. As a result, samples represented by $\mathbf{b}_i$ can also be represented by other basis vectors, which intensifies the differences in sample representations and reduces the consistency of representations.

---

> > ### Comment · Reviewer_SXo8 · 2025-08-04
> >
> > Thanks for the response. All my concerns has been addressed and I decide to improve my ratings.

---

> > > ### Author Response · Authors · 2025-08-04
> > > **Thanks for your positive feedback**
> > >
> > > We sincerely appreciate your valuable feedback and we will incorporate all suggestions to improve the final version.

---

### Official Review · Reviewer_gxAQ · 2025-06-22

**Clarity:** 3
**Significance:** 3
**Originality:** 3
**Rating:** 5
**Confidence:** 4

**Summary:**

This paper focuses on the unaligned multi-view clustering (UMC) problem and analyzes the shortcomings of existing UMC methods in fusing multi-view information. This paper points out that the alignment-based cross-view fusion strategies of those UMC methods may be unreliable since the alignment strategies fail to achieve ideal view-alignment results. Therefore, this paper provides an alignment-free consistency fusion framework, which eliminates the requirement for alignment strategies and directly extracts a consistent representation from each view to perform global cross-view fusion. Sufficient experiments demonstrate the excellent performance of the proposed model.

**Questions:**

Most of my questions are listed in the section above.  Additionally, for the ablation studies in Table 5, (b) conducts an ablation on the cross-view consistent basis space, where the loss constraint $ \mathcal{L}_{s} $ on the basis space seems to be ablated simultaneously, which is insufficient in evaluating the impact of the individual cross-view consistency basis space.

**Ethical Concerns:**

["NO or VERY MINOR ethics concerns only"]

**Final Justification:**

The authors propose a novel fusion framework for MVC, the rebuttal have addressed my score, so I decide to raise my score.

**Limitations:**

The model should be compared with the baseline models in 2025.

**Paper Formatting Concerns:**

There are no obvious formatting issues.

**Quality:**

3

**Strengths And Weaknesses:**

Strengths:
1. The analysis that alignment strategies fail to achieve ideal view-alignment results has been adequately explained in the Introduction and is understandable.
2. The idea of alignment-free cross-view fusion is interesting and the proposed alignment-free consistency fusion framework reasonably realizes this idea.
3. The experiments are sufficient. This paper evaluates the method on both small-scale and large-scale datasets, all of which show excellent clustering performance when compared with SOTA models.

---

> ### Author Rebuttal · Authors · 2025-07-29
>
> ************ **For Question #1:** ************
> --------------------------------------------------------------
>
> When the cross-view consistent basis space is ablated, the loss function $\mathcal{L}\_{s}$ on the basis space is indeed ablated as well. To evaluate the individual impact of the consistent basis space $\mathbf{B}$, we provide additional ablation studies in Table 1, where (1) ablates both $\mathcal{L}\_{s}$ and $\mathbf{B}$, and (2) only ablates $\mathcal{L}\_{s}$. From Table 1, (2) shows better performance than (1), demonstrating the effectiveness of cross-view consistent basis space in prompting autoencoders to extract consistent representations from each view.
>
> **Table 1**
> |         |   | ***Caltech7-5***   |        |  |                     | ***NoisyMNIST***  |       |           |           |
> |:---------:|:---------:|:------:|:------:|:------:|:--------:|:------:|:------:|:------:|:-----:|
> |         | Without (w/o) | **ACC**  | **NMI** | **PUR** | **ARI** | **ACC**  | **NMI** | **PUR** | **ARI** |
> | (1)     |   w/o $\mathcal{L}\_{s}$ & $\mathbf{B}$      | 0.7107   | 0.5989 | 0.7107 | 0.5312 | 0.4793   | 0.4136 | 0.5202 | 0.2862 |
> | (2)     | w/o $\mathcal{L}\_{s}$      | **0.8014** | **0.6983** | **0.8014** | **0.6406** | **0.5046** | **0.4573** | **0.5501** | **0.3297** |
>
> --------------------------------------------------------------
>
> ************ **For Limitation #1:** ************
> --------------------------------------------------------------
>
> The model should be compared with the baseline models in 2025.
>
> We provide an experimental comparison between our AF-UMC and the baseline model MGCCFF [1] (2025) in Table 2, where the incomplete rate is set to 0 and other parameters are set to the optimal values provided by the authors. Due to the excessive complexity of MGCCFF, Table 2 only shows the experimental comparison on small-scale datasets, with ACC as the metric. As shown in Table 2, AF-UMC outperforms MGCCFF on all small-scale datasets, demonstrating its excellent performance.
>
> **Table 2**
> |       |  ***Caltech7-5*** | ***Handwritten***   |   ***Scene***     | ***Caltech102-5*** |   ***Hdigit***    | ***Aloi***  |
> |:---------:|:---------:|:------:|:------:|:------:|:--------:|:--------:|
> |    MGCCFF  | 0.7857 | 0.8248 |  0.3713  | 0.1708 |  0.6167 | 0.5176 |
> | AF-UMC |   **0.8721**  |**0.9035**   |**0.4190** | **0.2275** | **0.6950** |**0.5399** |
>
> [1] Zhao, Liang, Ziyue Wang, Xiao Wang, Zhikui Chen, and Bo Xu. "Incomplete and Unpaired Multi-View Graph Clustering with Cross-View Feature Fusion." In Proceedings of the AAAI Conference on Artificial Intelligence, pp. 22786-22794. 2025.

---

> > ### Comment · Reviewer_gxAQ · 2025-08-04
> > **Response**
> >
> > The authors have addressed my concerns, I decide to raise my score.

---

> > > ### Author Response · Authors · 2025-08-04
> > > **Thank you for your comments**
> > >
> > > We sincerely appreciate your positive feedback and thoughtful comments. Your support is a great encouragement to our work.

---

### Official Review · Reviewer_UEME · 2025-06-24

**Clarity:** 3
**Significance:** 3
**Originality:** 3
**Rating:** 4
**Confidence:** 4

**Summary:**

This paper finds that current Unaligned Multi-view Clustering (UMC) methods often fail to achieve ideal view-alignment results, inevitably inducing their alignment-based cross-view fusion towards a biased direction. Different from current methods, this paper provides an alignment-free consistency fusion framework named AF-UMC, which first builds a cross-view consistent basis space and extracts consistent representations by projecting original samples of each view into this basis space, and then fuses cross-view consistent representations by pulling together these extracted representations at a global level, which realizes the alignment-free consistency fusion.

**Questions:**

1.	Why do the authors constrain the basis vectors to be linearly independent? Are there any negative effects if there exist linearly dependent basis vectors in the basis space?
2.	In my opinion, the cosine similarity function could quantify the similarity across basis vectors as their structural similarity. Why do the authors apply an additional exponential function to the obtained cosine similarity?
3.	In global contrastive fusion, each instance of fused cross-view consistent representation is encouraged to be closer to the global centers. Won’t this operation destroy the cluster distributions and make the distributions indistinguishable?

**Ethical Concerns:**

["NO or VERY MINOR ethics concerns only"]

**Final Justification:**

The rebuttal solves most of my concerns, and therefore I decide to maintain my positive score.

**Limitations:**

The authors only provide the visualizations of clustering results on small-scale datasets, but the visualizations on large-scale datasets are ignored. It is crucial to provide the visualizations on large-scale datasets in the manuscript to further verify the effectiveness of AF-UMC.

**Paper Formatting Concerns:**

I do not notice any major formatting issues in this paper.

**Quality:**

2

**Strengths And Weaknesses:**

***Strengths***
The work is well-motivated by analyzing the problem of the view-alignment operation existing in current UMC methods. Besides, the designed alignment-free consistency fusion framework is novel and effective in mitigating the degraded fusion performance caused by the undesired view-alignment operation. The extensive experimental results on various multi-view datasets demonstrate the superiority of the proposed AF-UMC.

***Weaknesses***
1.	The authors merely point out that current methods often fail to ideally align heterogeneous representations across views, but the specific process is not provided in this paper. The authors should introduce the alignment process of existing methods in detail and further note how the heterogeneous representations hinder the alignment process from achieving ideal view-alignment results.
2.	The cross-view consistent basis space is designed to participate in sample reconstructions of multiple views, where it seems inevitable to receive view-specific diversity from each view. I want to know whether the view-specific diversity will weaken the consistency within the basis space.
3.	For the ablation studies on loss functions of AF-UMC, an ablation study on $ \mathcal{L}_{r} $ is missing here.

---

> ### Author Rebuttal · Authors · 2025-07-29
>
> ************ **For Weakness #1:** ************
> ------------------------------------------------------------
>
>
> The alignment processes of existing methods are mainly divided into two categories: **Learning an alignment matrix $\mathbf{P}$** and **Employing the Hungarian algorithm**. Assuming there are two unaligned and heterogeneous representations $(\mathbf{H}^1, \mathbf{H}^2)$ from two views, and for a specific sample instance $k$, $((\mathbf{h}^1\_i)\_k, (\mathbf{h}^2\_j)\_k)$ denote its corresponding unaligned representations in $(\mathbf{H}^1, \mathbf{H}^2)$. (1) **Learning an alignment matrix $\mathbf{P}$:** These approaches assume that the consistent representations from each instance $k$ have a lower featural discrepancy, and attempt to learn an alignment matrix $\mathbf{P}$ by minimizing the discrepancy loss between $\mathbf{P}\mathbf{H}^1$ and $\mathbf{H}^2$. However, due to the inherent heterogeneity between $\mathbf{H}^1$ and $\mathbf{H}^2$, the representations from the same instance often exhibit a large featural discrepancy, making it difficult to align them by discrepancy minimization, thereby hindering the alignment process. (2) **Employing the Hungarian algorithm:** Unlike the above methods aligning representations by extra loss terms, these methods directly employ the existing Hungarian algorithm, which calculates the alignment costs of each possible pair $(\mathbf{h}^1_i, \mathbf{h}^2_j)$ from $\mathbf{H}^1=\\{\mathbf{h}^1_i\\}\_{i=1}^N$ to $\mathbf{H}^2=\\{\mathbf{h}^2_j\\}_{j=1}^N$ by a relevant cost function, e.g., Euclidean distance, cosine distance, etc, and then select the alignment result with the lowest total costs. However, due to the heterogeneity between $\mathbf{H}^1$ and $\mathbf{H}^2$, the alignment cost for some true corresponding pairs (e.g., $((\mathbf{h}^1_i)_k, (\mathbf{h}^2_j)_k)$) is often higher than that for some false corresponding pairs (e.g., $((\mathbf{h}^1\_i)\_k, (\mathbf{h}^2\_p)\_o), k \neq o$), where it is hard for Hungarian algorithm to select the true corresponding pairs for each instance, thereby hindering the alignment process.
>
> ------------------------------------------------------------
>
> ************ **For Weakness #2:** ************
> ------------------------------------------------------------
>
>
> The basis space only receives diversity at the beginning of model training. As training progresses, the diversity is gradually filtered out and will not weaken the consistency within the basis space. Specifically, the diversity only represents the information from a specific view, and has limited capacity in reconstructing samples from multiple views. Therefore, when the basis space is designed to reconstruct samples from multiple views, it tends to prioritize capturing cross-view consistency, facilitating the diverse information to fade over the model training. Besides, we constrain the basis space to be low-rank for capturing more compact consistency, which further reduces view-specific diversity.
>
> ------------------------------------------------------------
>
>
> ************ **For Weakness #3:** ************
> ------------------------------------------------------------
>
>
> Table 1 shows the ablation study for reconstruction loss $\mathcal{L}\_{r}$ on ***Caltech7-5*** and ***NoisyMNIST*** datasets, where (b) shows a significant performance improvement over (a), indicating that $\mathcal{L}_{r}$ plays a critical role in improving representation quality.
>
> **Table 1**
> |         | Loss functions      | ***Caltech7-5***  |    |  |                                | ***NoisyMNIST***|    |    |    |
> |:---------:|:---------------:|:--------:|:-------------:|:-----------:|:------:|:--------:|:------:|:------:|:------:|
> |         | $\mathcal{L}_{r}$   |          **ACC**  | **NMI** | **PUR** | **ARI** | **ACC**  | **NMI** | **PUR** | **ARI** |
> | (a)     |    | 0.5014   | 0.4366 | 0.5429 | 0.3056 | 0.4637   | 0.3924 | 0.5057 | 0.2826 |
> | (b)    | ✓ | **0.8721** | **0.7798** | **0.8721** | **0.7485** | **0.5899** | **0.4982** | **0.6247** | **0.4154** |
>
> ------------------------------------------------------------
>
> ************ **For Question #1:** ************
> ------------------------------------------------------------
>
>
> The linearly independent basis vectors have a characteristic that each of them cannot be linearly represented by others, which helps eliminate repeated consistent information (i.e., linearly dependent information) in the consistent basis space. If linearly dependent basis vectors exist, some consistent information will be repeatedly encoded across different basis vectors and samples with consistent information can be represented by different basis vectors, which intensifies the difference in sample representations and degrades the consistency of representations.
>
>
> ------------------------------------------------------------
>
> ************ **For Question #2:** ************
> ------------------------------------------------------------
>
>
> The exponential function $e^{(\cdot)/\tau_g}$ maps the value range $(-1, 1)$ of cosine similarity into a broader range $(0, e^{1/\tau_g})$, which amplifies structural differences among basis vectors, encouraging a clearer and more distinguishable similarity structure.
>
> ------------------------------------------------------------
>
> ************ **For Question #3:** ************
> ------------------------------------------------------------
>
> This operation will not destroy the cluster distribution and will not make the distribution indistinguishable. In addition to pulling instances toward global centers, the global contrastive fusion also introduces negative pairs to push dissimilar instances apart, preventing instances from being overly concentrated around the centers and preserving the distributional separability. Besides, Figure 3 provides the visualizations of the clustering results (see paper), where our AF-UMC shows a clear cluster distribution, confirming that this operation will not destroy the cluster distribution and will not make the distribution indistinguishable.
>
> ------------------------------------------------------------
>
> ************ **For Limitation #1:** ************
> ------------------------------------------------------------
>
>
> We perform the visualization comparison with other SOTA methods (FUMC, GCFAgg, LMVSC, MFLVC, OpVuC, SCMVC) on the large-scale dataset ***Cifar10***. Since we cannot directly provide a link to the visual results, we instead describe their distribution as follows: (1) FUMC, LMVSC, GCFAgg, MFLVC, OpVuC and SCMVC show a poor inter-class separation. (2) Our AF-UMC achieves well-separated clusters. **We will provide the visualization comparison in the revised manuscript.**

---

### Official Review · Reviewer_pBuo · 2025-07-02

**Clarity:** 2
**Significance:** 3
**Originality:** 3
**Rating:** 4
**Confidence:** 4

**Summary:**

The work explores unaligned multi-view clustering problem and propose that due to heterogeneity of various views, current alignment strategies are not fully effective. To address this issue, the authors proposed an alignment-free method, which builds a cross-view consistent basis space and extracts consistent representations from each view by projecting view-specific data onto the consistent basis space. Several experiments are conducted to support the efficacy of the method.

**Questions:**

1. Please give a more detailed explanation to demonstrate how the low-rank constraint helps to capture cross-view consistency into the basis space.
2. What are the justifications of setting the central view $core$ to the view with the largest original feature dimension?
3. What are the differences between the structurally equivalent basis vectors and the normal basis vectors?

**Ethical Concerns:**

["NO or VERY MINOR ethics concerns only"]

**Final Justification:**

The authors have addressed my primary questions, and by comprehensively considering the novelty and technical soundness, I will maintain my previous rating.

**Limitations:**

Yes.

**Paper Formatting Concerns:**

No obvious formatting issues.

**Quality:**

3

**Strengths And Weaknesses:**

##  Strengths:

1. The authors proposed a new problem of the ineffectiveness of existing alignment strategies due to view heterogeneity.
2. The authors provided a novel alignment-free solution by directly learning a consistent representation for global fusion;
3. Sufficient analyses and experiments are provided;

## Weaknesses:

1. Some details are not fully explained. Especially, the reason for the mismatched constructed basis spaces is not well analyzed.
2. It seems that the ablation study does not consider all the cases. Please conduct the missing cases or explain the potential reasons for not covering this case.
3. The clarity might need further improvement by specifying some sentences to make them clearer.

---

> ### Author Rebuttal · Authors · 2025-07-29
>
> ************ **For Weakness #1:** ************
> ---------------------------------------------------------
>
> Existing unaligned multi-view clustering methods fall into two main categories: feature-based methods and structure-based methods, whose reasons for the mismatched constructed basis spaces are as follows: (1) **Feature-based methods:** The basis space $\mathbf{Z}^v$ is constructed by intra-view sample reconstruction and only captures information from a single view $v$, as reflected by $\\|\mathbf{X}^v - \mathbf{H}^v\mathbf{Z}^v\\|\_{F}^{2}$. Due to the discrepancy in information across different views, it is unlikely for both basis vectors $(\mathbf{z}\_i^v, \mathbf{z}\_i^u)$ to coincidentally capture the matching information from different views $(v, u)$, respectively, thereby leading to mismatches across basis spaces $\\{\mathbf{Z}^v\\}\_{v=1}^V$. (2) **Structure-based methods:** As self-representation function $\\|\mathbf{X}^v - \mathbf{S}^v\mathbf{X}^v\\|\_{F}^{2}$, the basis space $\mathbf{X}^v$ (right) is directly constructed using the samples $\mathbf{X}^v$ (left) in each view $v$. Due to the misalignment across samples $\\{\mathbf{X}^v\\}_{v=1}^V$, the basis spaces constructed by different $\mathbf{X}^v$ are naturally mismatched across views.
>
> ---------------------------------------------------------
>
>
> ************ **For Weakness #2:** ************
> ---------------------------------------------------------
>
> Thank you for the suggestions. We supplement the missing cases as follows. **(1) Ablation study on reconstruction loss $\mathcal{L}_{r}$:** Table 1 shows the ablation study on $\mathcal{L}\_{r}$. From Table 1, (b) shows a significant performance improvement over (a), indicating that $\mathcal{L}\_{r}$ plays a critical role in improving representation quality. **(2) Ablation study on cross-view consistent basis space $\mathbf{B}$:** Considering that ablating $\mathbf{B}$ also removes the loss $\mathcal{L}\_{s}$ defined on $\mathbf{B}$, it is difficult to directly evaluate the impact of the individual basis space $\mathbf{B}$. To address this issue, we design the following ablation study in Table 2, where (a) ablates both $\mathcal{L}\_{s}$ and $\mathbf{B}$, and (b) only ablates $\mathcal{L}\_{s}$. From Table 2, (b) shows better performance than (a), demonstrating the effectiveness of cross-view consistent basis space in prompting autoencoders to extract consistent representations from each view.
>
> **Table 1**
> |         | Loss functions      | ***Caltech7-5***  |    |  |                                | ***NoisyMNIST***|    |    |    |
> |:---------:|:---------------:|:--------:|:------:|:------:|:------:|:--------:|:------:|:------:|:------:|
> |         | $\mathcal{L}\_{r}$   | **ACC**  | **NMI** | **PUR** | **ARI** | **ACC**  | **NMI** | **PUR** | **ARI** |
> | (a)     |    | 0.5014   | 0.4366 | 0.5429 | 0.3056 | 0.4637   | 0.3924 | 0.5057 | 0.2826 |
> | (b)    | ✓ | **0.8721** | **0.7798** | **0.8721** | **0.7485** | **0.5899** | **0.4982** | **0.6247** | **0.4154** |
>
> **Table 2**
> |         |   | ***Caltech7-5***   |        |  |                     | ***NoisyMNIST***  |       |           |           |
> |:---------:|:---------:|:----:|:--------:|:-------:|:--------:|:------:|:------:|:------:|:------:|
> |         | Without (w/o) | **ACC**  | **NMI** | **PUR** | **ARI** | **ACC**  | **NMI** | **PUR** | **ARI** |
> | (a)     |   w/o $\mathcal{L}_{s}$ & $\mathbf{B}$| 0.7107   | 0.5989 | 0.7107 | 0.5312 | 0.4793   | 0.4136 | 0.5202 | 0.2862 |
> | (b)     | w/o $\mathcal{L}_{s}$      | **0.8014** | **0.6983** | **0.8014** | **0.6406** | **0.5046** | **0.4573** | **0.5501** | **0.3297** |
>
> ---------------------------------------------------------
>
>
> ************ **For Weakness #3:** ************
> ---------------------------------------------------------
>
> We will revise the relevant sentences to make them clearer in the revised manuscript.
>
> ---------------------------------------------------------
>
> ************ **For Question #1:** ************
> ---------------------------------------------------------
>
> Low-rank constraint can reduce the diversity components from different views and promote the basis space to focus more on capturing cross-view consistency. Specifically, consider $V$ representations $\\{\mathbf{H}^v\\}\_{v=1}^V$ from $V$ views, where $\mathbf{H}^v \in \mathbb{R}^{N \times d}$. We vertically concatenate these representations into a matrix $\mathbf{H}^{all}=[\mathbf{H}^1; \cdots; \mathbf{H}^V] \in \mathbb{R}^{VN \times d}$, and then partition it into $K$ blocks according to the ground-truth class labels:
> $$\mathbf{M}=\begin{bmatrix}\mathbf{M}^1;
> \\ \cdots ;
> \\ \mathbf{M}^k;
> \\ \cdots ;
> \\ \mathbf{M}^K
> \end{bmatrix}
> $$
> where $\mathbf{M}^k \in \mathbb{R}^{N_k \times d}$ denotes the representations of the $N_k$ samples in class $k$. In each block $\mathbf{M}^k$, diversity components encourage the representations to be dissimilar across views and drive $\mathbf{M}^k$ to be high-rank, which indicates that diversity components tend to push $\mathbf{M}^k$ into a high-rank space since the diversity components cannot be effectively represented in a low-rank space. In contrast, consistency components encourage the representations to be similar across views within each block $\mathbf{M}^k$ and can be effectively represented in a low-rank space. Therefore, by appropriately lowering the rank of the basis space $\mathbf{B}$, the diversity components can be suppressed, enabling $\mathbf{B}$ to capture cross-view consistency more effectively.
>
> ---------------------------------------------------------
>
> ************ **For Question #2:** ************
> ---------------------------------------------------------
>
> In multi-view datasets, lower-dimensional views often provide a more limited and fragmented description of the samples, and the representations extracted from these views usually exhibit a biased or unstable global center. Consequently, these views are less suitable as the central view for cross-view global fusion. Instead, the view with the largest original feature dimension provides a more comprehensive description of the samples, promoting the extracted representations with a more stable global center. Therefore, the view with the largest original feature dimension is more suitable as the central view for cross-view global fusion.
>
> ---------------------------------------------------------
>
> ************ **For Question #3:** ************
> ---------------------------------------------------------
>
> There are two differences between the structurally equivalent basis vectors and the normal basis vectors. (1) **Intrinsic property:** Structurally equivalent basis vectors can be interchanged without changing the graph structure [1], while normal basis vectors cannot. Specifically, consider a graph $G^1=(\mathbf{Z}^1, \mathbf{S}^1)$, where $\mathbf{Z}^1$ denotes the vertices (i.e., basis vectors) and $\mathbf{S}^1$ denotes the graph structure. Structurally equivalent basis vectors $(\mathbf{z}_i^1, \mathbf{z}_j^1)$ can be interchanged in $\mathbf{Z}^1$ without changing $\mathbf{S}^1$, while normal basis vectors cannot [1]. (2) **Influence on structure matching:** When matching $G^1$ with another graph $G^2=(\mathbf{Z}^2, \mathbf{S}^2)$ using a structure-based matching function $\varPhi(\mathbf{S}^1, \mathbf{S}^2)$, multiple matching results $\\{\mathbf{Z}^1 \xrightarrow{m} \mathbf{Z}^2\\}$ arise by interchanging structurally equivalent $(\mathbf{z}_i^1, \mathbf{z}_j^1)$ without changing the value of $\varPhi(\mathbf{S}^1, \mathbf{S}^2)$. In this case, structurally equivalent basis vectors hinder the ideal and unique matching between $\mathbf{Z}^1$ and $\mathbf{Z}^2$, whereas normal basis vectors do not have this influence.
>
> [1] Yang, Dominic, Yurun Ge, Thien Nguyen, Denali Molitor, Jacob D. Moorman, and Andrea L. Bertozzi. "Structural equivalence in subgraph matching." IEEE Transactions on Network Science and Engineering 10, no. 4 (2023): 1846-1862.

---

> > ### Comment · Reviewer_pBuo · 2025-08-09
> >
> > The authors have addressed my primary questions, and the work indicates its novelty and contributions.  I will maintain my previous rating.

---

### Decision · Program_Chairs · 2025-09-17

**Decision:**

Accept (poster)

**Comment:**

This paper proposes AF-UMC, a novel alignment-free fusion framework for unaligned multi-view clustering. Unlike existing methods that rely on complex and often unreliable alignment strategies to match samples across views before fusion, AF-UMC bypasses explicit alignment entirely. Instead, it directly extracts consistent representations from each view by projecting view-specific data onto a shared, low-rank, cross-view consistent basis space, which is constructed using a cross-view reconstruction loss and a structural clarity regularization to avoid learning redundant basis vectors. These representations are then globally fused using an Instance global contrastive enhancement mechanism, which pulls the global centers of views together and enhances each instance's consistency via contrastive learning. The method significantly reduces computational complexity and improves clustering performance by avoiding alignment errors. Experiments on ten datasets show state-of-the-art results.

I have checked the author-reviewer discussions, and all the reviewers acknowledged the rebuttal and replied to the authors. This paper finally receives the scores of 4, 4, 5, 5, which means all the reviewers would like to be on the positive side.

This paper is well-motivated, which addresses a fundamental limitation in unaligned multi-view clustering: the reliance on error-prone alignment steps. The proposed AF-UMC framework is novel, efficient, and empirically validated across a wide range of datasets.

Based on the above situations, I vote for acceptance.